# Regulation of the SIRT3/SOD2 Signaling Pathway by a Compound Mixture from *Polygonum orientale* L. for Myocardial Damage

**DOI:** 10.3390/ph17101288

**Published:** 2024-09-27

**Authors:** Chunhua Liu, Yu He, Mingjin Wang, Jia Sun, Jie Pan, Ting Liu, Yueting Li, Meng Zhou, Yong Huang, Yongjun Li, Yanmin Zhang, Yuan Lu

**Affiliations:** 1State Key Laboratory of Functions and Applications of Medicinal Plants, Engineering Research Center for the Development and Application of Ethnic Medicine and TCM (Ministry of Education), Guizhou Medical University, Guiyang 550004, China; 2School of Pharmacy, Guizhou Medical University, Guiyang 550025, China; 3Guizhou Provincial Key Laboratory of Pharmaceutics, Guizhou Medical University, Guiyang 550004, China; 4Laboratory of Molecular Design and Drug Discovery, School of Science, China Pharmaceutical University, 639 Longmian Avenue, Nanjing 211198, China

**Keywords:** *Polygonum orientale* L. active components, myocardial ischemia, network pharmacology, H9c2, oxidative stress, SIRT3/SOD2 signal pathway

## Abstract

Background: *Polygonum orientale* L. (PO) has demonstrated notable efficacy in treating coronary heart disease. Previous research identified eight key active components in PO for cardiomyocyte protection, but the underlying mechanisms remained unclear; Methods: Network pharmacology and molecular docking were used to identify potential target proteins of PO’s active components. Experimental models assessed the cardioprotective effects and mechanisms; Results: Network analysis and molecular docking revealed that the active components exhibited the highest binding affinity with SOD2, indicating it as a key element in the cardiac protection of PO. In vivo, PO extract improved myocardial structure and function, and increased SOD2 protein levels. In vitro, the active components of PO (Mixture) mitigated oxidative stress and apoptosis, upregulating SIRT3 and decreasing acetylated SOD2, leading to increased SOD2 and reduced ROS levels. The observed effects were reversed by a SIRT3 inhibitor, indicating the involvement of the SIRT3/SOD2 signaling pathway; Conclusions: This comprehensive approach elucidated the critical mechanisms underlying the cardioprotective properties of PO’s bioactive constituents, highlighting the regulation of the SIRT3/SOD2 signaling pathway as a new mechanism for PO’s anti-cardiovascular disease effects, and suggesting the Mixture’s potential as a promising drug candidate.

## 1. Introduction

Cardiovascular disease (CVD) has long posed a threat to human health. According to the “China Cardiovascular Disease Report 2022” [1], CVD incidence and mortality rates in China continue to show an upward trend. The number of individuals currently suffering from cardiovascular diseases is estimated to be around 330 million, making it the leading cause of mortality among both urban and rural residents in the country. Among CVDs, myocardial ischemia (MI) is a prevalent and causative factor in clinical practice. Prolonged myocardial ischemia leads to irreversible damage to myocardial tissue and contributes to the development of other critical conditions, including myocardial infarction, coronary heart disease, and heart failure. Myocardial ischemia represents a pathological state resulting from diminished blood flow and reduced oxygen supply to the heart, driven by a variety of factors [2]. The intricate pathogenesis of myocardial ischemia encompasses aspects such as disruptions in energy metabolism, cellular apoptosis, oxidative stress, inflammatory responses, and intracellular calcium overload [3]. In recent years, changes in lifestyle have contributed to an annual increase in the incidence and mortality rates of myocardial ischemic diseases, posing a significant threat to human life and health, and impacting the overall quality of life. Traditional Chinese medicine (TCM), characterized by its multi-component, multi-target, and multi-pathway synergistic effects in disease treatment, offers a promising avenue for addressing complex chronic conditions [4]. Given its minimal side effects, exploring TCM for potential treatments for myocardial ischemia could provide valuable insights for new drug development.

*Polygonum orientale* L. (PO) is derived from the dried fruit spikes and leafy stems of the Polygonaceae family, the botanical resource of which is widely distributed across Guizhou province, China. Traditionally, it possesses therapeutic attributes such as heat-clearing, detoxification, wind-dispersing, dampness-removing, blood-activating, and swelling-reducing effects, commonly used for treating cardiovascular conditions, including coronary heart disease and chest discomfort [5]. Furthermore, our previous study demonstrated that PO exhibits significant therapeutic benefits for cardiovascular diseases [6]. Several formulations that primarily consist of PO, such as “Hongyexintong Soft Capsules” and “Injectable Hongcao Lyophilized Powder”, were developed by our team, with considerable efficacy in the management of myocardial infarction [7,8]. For example, the latter formulation has exhibited the capacity to enhance myocardial oxygen supply, reduce infarct size, and lower serum LDH and CK levels, thus substantiating its cardioprotective potential [9]. Similarly, “Hongyexintong Soft Capsules” have proven effective in minimizing myocardial infarct size [8]. The efficacy of these formulations in the treatment of angina pectoris and coronary heart disease has been well-established in clinical practice, with minimal adverse effects.

PO contains a diverse array of compounds, including flavonoids and phenolic acids [10], with various pharmacological effects that span anti-tumor, anti-myocardial ischemia, anti-hypoxia, and anti-inflammatory properties [11,12]. Drawing upon investigations on PO conducted by our research group, including chemical composition, active fractions, extraction processes, and pharmacological activities and process in vivo [13], compounds such as flavonoids, phenolic acids, and phenylpropanoids have been identified as the pharmacologically active constituents of PO, including orientin, isoorientin, vitexin, kaempferol-3-O-β-d-glucoside, quercetin, *N*-*trans*-feruloyltyramine, paprazine, and protocatechuic acid. Nevertheless, the molecular mechanisms of these active ingredients remain elusive, presenting a significant impediment to the further development and utilization of PO. Consequently, the investigation of the active constituents and their mechanisms of action in the context of myocardial ischemia treatment is of great significance.

Due to the complexity of the chemical components and diversity of biological activities of TCM, with a multi-component and multi-target characteristic, elucidating the molecular mechanism of TCM has always been a bottleneck in modern research of TCM. Network pharmacology, based on the “disease-gene-target-drug” interaction network, studies the effects of drugs on diseases from a systemic and holistic perspective [14]. Moreover, it emphasizes that the process of drug action in the body is a complex network, a mechanism with a “multi-component, multi-target, multi-pathway”, which aligns well with the holistic philosophy of TCM and its principle of syndrome differentiation and treatment. With the advancement of systems biology and bioinformatics, methods such as network pharmacology and molecular docking have been widely applied in the research on the pharmacological basis and mechanism of action of TCM [15]. These methods have gained increasing recognition from scientists working on TCM research, providing new insights into exploration of molecular mechanisms of TCM and even TCM’s active constituents.

Thereby, integrated techniques encompassing network pharmacology and molecular docking were employed to predict the potential targets through which the bioactive constituents of PO exert cardioprotective effects. Combined with both in vivo and in vitro models, the cardioprotective efficacy and underlying mechanisms of PO and the active ingredients were validated. Collectively, these findings could offer a solid foundation for the advanced therapeutic development and utilization of PO.

## 2. Results

### 2.1. Acquisition of the Active Components of PO and Their Potential Targets for Treating Cardiovascular Diseases

Based on the preliminary laboratory research on the chemical composition, pharmacological effects, and serum drug chemistry of PO, the active components in PO are tentatively identified as orientin, isoorientin, vitexin, quercetin, kaempferol-3-O-β-d-glucoside and protocatechuic acid [16]. Previous studies have indicated the presence of phenylpropanoid components in PO, exhibiting favorable biological activity [17]. Therefore, *N*-*trans*-feruloyltyramine and paprazine are considered as active components of PO. Collectively, the confirmed active component group of PO comprises orientin, isoorientin, vitexin, kaempferol-3-O-β-d-glucoside, quercetin, *N*-*trans*-feruloyltyramine, paprazine and protocatechuic acid, with their structures presented in Figure 1A.

Using the above eight active ingredients as keywords, target retrieval was conducted in databases such as TCMSP and SwissTargetPrediction. Data were integrated, resulting in a total of 174 potential target genes. Cardiovascular disease-related target genes were searched from databases including OMIM, Gene Cards, Drugbank, and DisGeNET. After eliminating duplicate target genes, a total of 7928 disease-related proteins were obtained, as shown in Figure 1B. The potential targets of the eight components were compared with disease-related targets using the VLOOKUP function in Excel (version no. 14.0.7261), revealing 98 common targets.

### 2.2. Construction of PPI Network

The 98 shared targets were imported into the STRING platform to retrieve protein-protein interaction relationships. Visualization was conducted using Cytoscape 3.7.2 software, as shown in Figure 1C. After removing free genes, the resulting network comprises 97 nodes and 468 edges, where nodes represent targets and edges denote the interaction relationships between targets. The greater the number of edges, the stronger the correlation between proteins. The degree value indicates the node’s significance, with a higher degree value signifying greater importance of the target. The top 20 key functional targets, ranked by degree value in descending order, include VEGFA, CASP3, EAR1, MAPK8, PTGS2, HSP90AA1, IL1B, CCL2, AR, RHOA, PPARG, MAPK14, MAP2K1, MMP2, SOD2, KDR, SERPINE1, HSP0AB1, PLAU, and AHR.

### 2.3. Molecular Docking Analysis

Docking was performed for the eight active constituents and the top twenty core targets. Among them, the binding scores between the eight constituents and AR, PPARG, and SOD2 were relatively low, as shown in Table 1. Oxidative damage plays a crucial role in the pathology of cardiovascular diseases [18], and there was evidence that PO and/or these compounds demonstrated significant antioxidant effects in their treatment [19]. As SOD2 is a key enzyme in the body’s antioxidant system, in this study, SOD2 was selected as a focal point for further investigation. The binding modes of the compounds to SOD2 are shown in Figure 1D.

### 2.4. PO Extract Alleviated the Myocardial Damage Caused by LAD Ligation

The mouse electrocardiogram was monitored using a biological function experiment system, and further analysis was conducted on the changes in the ST segment amplitude. The results indicated a significant shortening of the ST segment amplitude in the MI group compared to the Sham group (*p* < 0.001). When compared to the MI group, both the POE group and MTT group exhibited a pronounced increase in the ST segment amplitude (*p* < 0.001). To confirm the extent of ischemia, heart tissues were sectioned and subsequently stained in a 1% TTC solution under light-avoiding conditions for the quantification of the myocardial infarction area. The findings demonstrated a significant elevation in the myocardial infarction area in the MI group compared to the Sham group (*p* < 0.001). In comparison to the MI group, both the POE group and MTT group showed a substantial reduction in the myocardial infarction area (*p* < 0.01), as detailed in Table 2.

Subsequently, the damage to ischemic heart tissues was assessed through H&E staining. As depicted in Figure 2A, the Sham group exhibited normal myocardial tissue structures with well-organized and closely arranged cardiomyocytes. Fibrosis and inflammatory cell infiltration were not observed, indicating the absence of evident lesions. In contrast, the MI group revealed severe abnormalities in their myocardial tissue structure, characterized by the disorganized arrangement of cardiomyocytes and extensive myocardial tissue fibrosis. The POE group and MTT group demonstrated a noticeable reduction in the severity of myocardial injury. The myocardial tissue structure appeared to be mostly normal, with minimal cardiomyocyte degeneration, a relatively orderly arrangement, and some enlargement of the myocardial interstitium. Furthermore, CK, a myocardial enzyme reflecting the degree of myocardial damage, was measured. Our results indicated that POE significantly reduced the level of CK (Figure 2B). BNP, a marker for assessing cardiac function, can be utilized to monitor ischemic damage following myocardial infarction. Our results indicated that compared to the Sham group, the MI group exhibited a significant elevation in BNP levels (*p* < 0.001). In comparison to the MI group, the POE significantly reduced BNP levels (*p* < 0.01) (Figure 2C). Collectively, POE demonstrated a remarkable cardioprotective effect in ameliorating the myocardial damage induced by LAD ligation.

### 2.5. POE Enhances the Expression of SOD2 and the Activity of SOD, Alleviating Apoptosis

Based on the molecular docking results, we further assessed the impact of POE on the protein expression of SOD2. The validation results using the Western blot demonstrated that in the myocardial ischemia group, the SOD2 level was significantly reduced compared to the Sham group. Following treatment with POE, the SOD2 level exhibited a marked increase, indicating that POE has the capability to elevate the SOD2 level in myocardial ischemic mice (Figure 2D,I). To assess the influence of POE on the activity of SOD, the activity was measured using a SOD assay kit. The results indicated a significant increase in SOD activity with POE treatment (Figure 2H).

In order to further evaluate the protective effects of POE on the heart, we examined the expression of apoptosis-related proteins. Western blot results (Figure 2D–G) revealed a significant upregulation in the expression of apoptosis-related proteins Bax, Caspase 3, and Caspase 9 in the model group compared to the Sham group (*p* < 0.001). In contrast, the expression of these proteins was markedly reduced in the POE group relative to the model group. This indicates that myocardial ischemic disease induces apoptosis in cardiac cells, and POE has the potential to ameliorate apoptosis caused by myocardial ischemia.

### 2.6. The Active Component Mixture from PO Effectively Protects H9c2 Cardiomyocytes from Injury Caused by H_2_O_2_

Given the pivotal role of oxidative stress in the pathogenesis of cardiovascular diseases [20], an H_2_O_2_-induced oxidative damage model in H9c2 cardiomyocytes was utilized to further investigate the protective effects of active components from PO. Initially, the effect of eight PO’s active components on H9c2 cells was investigated at various concentrations to determine safe levels for each (Appendix A). After assessing H_2_O_2’_s injurious impact on H9c2 cells across different concentrations, we established the optimal condition for modeling at 400 μmol/L (Appendix A). Simultaneously, the protective influence was examined of varying POE concentrations against H_2_O_2_-induced damage, identifying 80 μg/mL as the most effective dose (Appendix A). Therefore, subsequent studies were conducted at this concentration. Following treatment with a mixture of active components (Mix) and POE, H9c2 cell viability significantly improved, mitigating the reduction caused by H_2_O_2_ (Figure 3A). Additionally, LDH levels in the supernatant of H9c2 cells were measured, revealing a marked increase in LDH release in cells treated with H_2_O_2_ compared to control cells (*p* < 0.001). Notably, the Mix group exhibited a substantial decrease in LDH release relative to the H_2_O_2_ group (*p* < 0.001) (Figure 3B). These findings collectively suggest that PO’s active component mixture notably protects cardiomyocytes from H_2_O_2_-induced damage.

### 2.7. The Active Component Mixture from PO Boosts the Antioxidant Capabilities and Mitochondrial Membrane Potential in Myocardial Cells

The intracellular activities of superoxide dismutase (SOD) and catalase (CAT), crucial antioxidant enzymes, were evaluated in H9c2 cells, as illustrated in Figure 3C,D. Notably, the enzymatic activities of SOD and CAT were substantially reduced in H9c2 cells exposed to H_2_O_2_ compared to normal cells (*p* < 0.001). In contrast, both the Mix group and the POE group demonstrated significant enhancements in SOD and CAT activities relative to the H_2_O_2_ group (*p* < 0.05, *p* < 0.01, *p* < 0.001). Concurrently, the accumulation of intracellular ROS was measured, revealing a marked increase in ROS in H_2_O_2_-treated H9c2 cells compared with normal cells (*p* < 0.001). Pre-treatment with the Mix and POE resulted in a significant reduction in ROS levels (*p* < 0.001) (Figure 3E). Oxidative damage in cells typically leads to a reduction in mitochondrial membrane potential. This potential change in H9c2 cells was assessed using the JC-1 reagent kit. The findings, presented in Figure 3F,G, demonstrate a significant decrease in mitochondrial membrane potential in H9c2 cells exposed to H_2_O_2_ compared to normal cells (*p* < 0.001). Notably, pre-treatment of H9c2 cells with Mix and POE resulted in a substantial elevation of the mitochondrial membrane potential (*p* < 0.001), suggesting that Mix and POE effectively counteract the mitochondrial membrane potential reduction induced by H_2_O_2_. Collectively, pretreatment with Mix and POE significantly enhances the antioxidant defenses of H9c2 cells against H_2_O_2_-induced injury.

### 2.8. The Active Component Mixture from PO Mitigates Cell Apoptosis Triggered by H_2_O_2_

To further assess the protective effect of Mix on myocardial cells, the apoptosis rate and the expression of apoptosis-related proteins were examined. As shown in Figure 4A,B, the apoptosis rate of H9c2 cells following H_2_O_2_ exposure significantly increased compared to normal cells (*p* < 0.001), while pretreatment of H9c2 cells with Mix significantly reduced this rate (*p* < 0.001), suggesting its efficacy in reducing H_2_O_2_-induced apoptosis in H9c2 cells. Additionally, Western blot results (Figure 4C–F) demonstrated that protein expression levels of Bax, Caspase 3, and Caspase 9 in H9c2 cells treated with H_2_O_2_ were significantly higher than those in the normal group (*p* < 0.05, *p* < 0.01, *p* < 0.001). In contrast, Mix significantly decreased the protein expression levels of Bax, Caspase 3, and Caspase 9 (*p* < 0.05, *p* < 0.01), providing further evidence of its ability to mitigate apoptosis induced by H_2_O_2_ in H9c2 cells.

### 2.9. The Role of the Active Component Mixture from PO in Modulating the SIRT3/SOD2 Signaling Pathway

SIRT3, a mitochondrial protein deacetylase, plays a crucial role in regulating SOD2 activity through deacetylation, thereby impacting the cellular capacity to eliminate intracellular superoxide radicals [21]. Consequently, we hypothesized whether the Mix could modulate SIRT3, subsequently influencing the activity of SOD2. To explore this, the protein expression of SIRT3, SOD2, and AC-SOD2 was assessed in H9c2 cells exposed to H_2_O_2_ and treated with Mix. It was found that H_2_O_2_ stimulation led to a reduction in the expression of SIRT3 and SOD2, accompanied by an increase in the AC-SOD2/SOD2 ratio in H9c2 cells (Figure 5A–D). However, these H_2_O_2_-induced alterations were mitigated by pretreatment with the Mix, suggesting a potential regulatory role of PO’s active components in the SIRT3/SOD2 signaling pathway.

To further assess the involvement of SIRT3 in the antioxidant stress response mediated by the Mix, the SIRT3 selective inhibitor 3-TYP was employed to treat H9c2 cells. As illustrated in Figure 5E–H, Mix increased the protein levels of SIRT3 and SOD2, while decreasing the ratio of AC-SOD2/SOD2 induced by H_2_O_2_. This effect, however, was reversed by the SIRT3 inhibitor 3-TYP. RT-PCR analysis (Figure 5I) demonstrated that after exposure to H_2_O_2_, there was a decline in SOD2 transcription in H9c2 cells, while Mix elevated SOD2 transcription levels. Notably, this elevation was similarly reversed by 3-TYP, which underscores that the enhancement of SOD2 expression by Mix occurs through the modulation of the SIRT3 protein.

To further validate SIRT3’s involvement in the protective effects of the Mix on myocardial cells, we examined the influence of a SIRT3 inhibitor on Mix’s ability to eliminate ROS and regulate apoptosis-related protein expression. The results depicted in Figure 6A revealed a significant increase in ROS levels in H9c2 cells stimulated by H_2_O_2_, and pretreatment with Mix mitigated the increased ROS levels. However, the SIRT3-specific inhibitor counteracted this reduction, thereby impacting its antioxidative response, which suggests PO’s antioxidative action is modulated by SIRT3. As illustrated in Figure 6B–E, Mix markedly diminished the protein levels of apoptosis-related proteins, including Bax, Caspase 3, and Caspase 9, an effect that was negated by 3-TYP. Collectively, these findings suggest that Mix exerts a protective role in myocardial cells by modulating the SIRT3/SOD2 signaling pathway. To further illustrate these key points, the following image (Figure 7) provides a compelling visual summary that encapsulates the molecular mechanism of Mix for treatment of myocardial damage.

## 3. Discussion

Cardiovascular disease is the leading cause of non-communicable disease mortality, with both its incidence and death rates steadily rising in recent years. The urgent priority is to identify effective methods and medications for preventing and treating cardiovascular diseases. TCM offers unique advantages in treating chronic diseases, making it a valuable resource for developing medications for cardiovascular diseases. However, the unclear mechanisms of action and the identification of primary active components in TCM have posed a significant bottleneck in modern research. The use of interdisciplinary techniques, such as network pharmacology, molecular docking, and in vivo and in vitro experimental validation, is a vital strategy for tackling the complex issues associated with TCM. Therefore, based on previous research, this study selected eight active components from PO. Utilizing network pharmacology, the potential target proteins of these active components were investigated, and molecular docking techniques further identified key proteins, revealing that SOD2 could be a key target regulated by these active components.

The pathogenesis of myocardial ischemia is complex, involving various factors that can lead to the occurrence of ischemic heart disease. These factors primarily include disorders in energy metabolism, apoptosis, oxidative stress, inflammatory responses, and intracellular calcium overload [22]. Among them, oxidative stress is mainly attributed to the excessive generation of oxygen-free radicals, playing a crucial role in the development of myocardial ischemic diseases. Oxidative stress refers to an imbalance between the body’s antioxidant system and intracellular reactive oxygen species (ROS) or reactive nitrogen species (RNS). This imbalance leads to the accumulation of active molecules, causing oxidative damage processes such as lipid peroxidation, DNA oxidation, and protein glycation [23].

SOD is one of the essential antioxidant enzymes in the body, responsible for eliminating oxygen-free radicals and protecting cells from oxidative stress damage [24]. SOD is also a metal enzyme with a catalytic center that contains a metal ion. Based on the different metal ions, it can be classified into four types [25]: ① Cu/Zn-SOD, mainly distributed in the cytoplasm, mitochondria, cell nucleus, and bacterial cytoplasm of eukaryotes; ② Mn-SOD, mainly distributed in the mitochondria of prokaryotes and eukaryotes; ③ Fe-SOD, found in prokaryotes and certain organisms such as archaea, obligate anaerobic bacteria, and facultative aerobic bacteria; ④ Ni-SOD, discovered in actinomycetes and cyanobacteria, primarily located in the cytoplasm of actinomycetes and blue-green algae.

Mn-SOD, also known as SOD2, plays a crucial role in balancing intracellular ROS, and its high activity can protect organisms from oxidative stress damage [26]. In conditions like myocardial ischemia or myocardial infarction, changes in SOD2 activity are closely associated with myocardial protection [27]. Moderate SOD2 activity helps alleviate myocardial oxidative stress, reduce damage to myocardial cells, and protect heart health [27]. Therefore, we validated the effects of POE and Mix on SOD2 protein expression and SOD activity in both animal and cell models. Our research results indicate that PO and Mix can significantly increase SOD2 protein levels and enhance SOD activity.

SIRT3, an NAD+-dependent deacetylase, plays a pivotal role in myocardial ischemia-reperfusion disease by deacetylating proteins such as FOXO3 and SOD2 [28]. SOD2, regulated through reversible lysine acetylation by SIRT3, is crucial in this network. Research by Ma et al. showed that preconditioning, such as lateral aortic constriction, activates SIRT3/SOD2-dependent pathways, which can mitigate myocardial autophagic cell death [29]. The loss of SIRT3 leads to increased SOD2 acetylation, causing severe oxidative stress, hypertension, and endothelial dysfunction [30]. Recent studies indicate that modulating the SIRT3/SOD2 pathway can reduce oxidative stress and apoptosis in myocardial ischemia-reperfusion, thereby protecting myocardial cells [28,29,31]. This highlights the vital role of SOD2 in ischemic heart disease therapy, governed by SIRT3 regulation. Our study thus delves into the effect of Mix on the SIRT3/SOD2 pathway. The results reveal that Mix amplifies the reduction of SIRT3 and SOD2 caused by H_2_O_2_, reduces the elevated AC-SOD2/SOD2 ratio, and this effect is counteracted by the SIRT3 inhibitor 3-TYP. Furthermore, 3-TYP impedes Mix’s ability to reduce ROS levels and the expression of apoptosis-related proteins. These findings collectively suggest that Mix contributes to myocardial cell protection by influencing the SIRT3/SOD2 pathway, demonstrating its therapeutic potential for cardiovascular diseases.

While this study has partially elucidated the material basis and mechanism of action of PO in treating cardiovascular diseases, the combination of active components is based solely on the content in the extract, which may not represent the optimal combination ratio. Further optimization of the combination formula is necessary to advance the development of new drugs based on the active component mixture of PO. Moreover, the validation of the mechanism in this study, mainly based on cell models, has inherent limitations. Consequently, additional verification using genetically modified mice and other sophisticated techniques is of great importance to solidify the results.

## 4. Materials and Methods

### 4.1. Reagents and Antibodies

Tribromoethanol (C11707118) was purchased from Shanghai McLean Biochemical Technology Co., Ltd. (Shanghai, China). Tertiary amyl alcohol (F2004116) was obtained from Shanghai Aladdin Biochemical Technology Co., Ltd. (Shanghai, China). Metoprolol tartrate (181115) was sourced from Yantai Juxian Pharmaceutical Co., Ltd. (Yantai, China). Thirty percent hydrogen peroxide (20180615) was purchased from Sinopsin Chemical Reagent Effective Company (Beijing, China). Orientin (AF9052413), isoorientin (AF20051551), vitexin (AF8111891), kaempferin-3-O-β-d-glucoside (AF8062705), *N*-*trans*-feruloyltyramine (AF20060301), protocatechuic acid (AF6121206), and quercetin (AF20032451) with a purity ≥ 98% were all purchased from Chengdu Alfa Biotechnology Co., Ltd. (Chengdu, China). High-glucose Dulbecco’s Modified Eagle Medium (8121235), Australian fetal bovine serum (2167759CP), 0.25% trypsin, and dual antibiotics (2199839) were acquired from Gibco (Grand Island, NY, USA). Creatine kinase CK (20201211), brain natriuretic peptide BNP (20201221), superoxide dismutase SOD (20201217), and LDH (Lot. 20210510) kits were provided from Nanjing Jiancheng Bioengineering Institute (Nanjing, China). SOD (NO. 02821210418), CAT (NO. 031021210422), ROS detection (NO. 011521210429), and mitochondrial membrane potential detection (NO. 032421210414) kits were obtained from Shanghai Biyuntian Biological Co., Ltd. (Shanghai, China). Anti-Bax (HN1221) antibody was acquired from Hangzhou Hua’an Biological Co., Ltd. (Hangzhou, China). The CCK-8 kit (NO. K10181233EF5E) was sourced from APE Inc. (Elk Grove Village, IL, USA). The apoptosis kit (0252058) was from BD (Franklin Lakes, NJ, USA). Anti-Caspase 3 (9662S), anti-Caspase 9 (9504T), and anti-SIRT3 (C73E3) antibodies were purchased from CST Corporation (Houston, TX, USA). The anti-GAPDH (GR200347-40) and anti-AC-SOD2 (GR3037648-15) antibodies were obtained from Abcam (Cambridge, UK). Anti-SOD2 antibody (00067140) and goat Anti-Rabbit (20000217) second antibodies were purchased from Proteintech Inc. (Rosemont, IL, USA). The SIRT3 selective inhibitor 3-TYP (100616) was acquired from MCE Corporation (Monmouth Junction, NJ, USA). The total RNA extraction kit (0000484334) was sourced from Promeg (Beijing) Biotechnology Co., Ltd. (Beijing, China). The reverse transcription kit (AL50947A) and TB GreenR Premix EX TaqTM II (AL61811A) were supplied from Takara Corporation (Kusatu, Japan).

PO (No. 20191124) was harvested from Panzhou city, Guizhou Province, which was authenticated by associate professor Liu Chunhua from Guizhou Medical University, and the specimen (HC-1) is housed in the Traditional Chinese Medicine Specimen Museum of Guizhou Medical University School of Pharmacy.

### 4.2. Prediction of Drug Targets and the Construction and Analysis of Protein-Protein Interaction (PPI) Networks Involving Target Proteins

Based on previous experimental research, a search was conducted in databases such as TCMSP and SwissTargetPrediction using the search terms “orientin”, “isoorientin”, “vitexin”, “kaempferol-3-O-β-d-glucoside”, “quercetin”, “*N*-*trans*-feruloyltyramine”, “paprazine”, and “protocatechuic-acid” to retrieve information related to the targets of these bioactive compounds. Furthermore, using the keywords “Myocardial ischemia” and “Coronary Artery Disease”, disease-related targets were searched for in the OMIM, Gene Cards, Drugbank, and DisGeNET databases. The common targets between the active compounds and heart ischemic diseases were obtained using the VLOOKUP function in Microsoft Excel 2010. A relationship network of “active compounds-targets-myocardial ischemia” was constructed utilizing Cytoscape v3.7.2 software (https://cytoscape.org/).

The shared target protein information was imported into the String database (https://string-db.org), a resource for exploring known and predicted protein-protein interactions [32]. The species was specified as “Homo sapiens”, and a confidence threshold was set at 0.400 to generate a Protein-Protein Interaction (PPI) network. Data visualization was performed using Cytoscape 3.7.2 software, and the “Network Analysis” function was utilized to conduct a topological analysis of the PPI network. In the network, node size represents the degree value, with larger nodes indicating higher degree values. These degree values reflect the extent of protein interactions and were used to select core proteins.

### 4.3. Molecular Docking

Eight bioactive compounds were docked with the top 20 key target proteins, ranked by degree value from PPI network analysis, using the AutoDock 4.2.6 [33]. The structures of these small molecules were prepared by the AutoDockTools, while the crystallographic structures corresponding to the 20 potential target proteins were retrieved from the Protein Data Bank (PDB, https://www.rcsb.org/). These protein structures were then meticulously prepared using Pymol (https://pymol.org) [34] and AutoDockTools, which involved retaining a single protein chain, removing all non-essential ions, and stripping water molecules located beyond 5 Å from the binding sites, followed by adding hydrogens and charges. Docking grids were centered on the ligand present in the protein crystal structure using the AutoGrid4 module, and subsequent molecular docking was performed utilizing the AutoDock4 module by setting the searching algorithm to be a genetic algorithm. For each molecule, 10 conformations were kept, and the best one was picked out for further analysis.

### 4.4. Preparation of P. orientale Extract (POE)

The preparation process of PO extract was as described previously [6]. Briefly, dried PO materials of 5 kg were taken and extracted three times with a ten-fold volume of water for 1 h each time. After each extraction, the solution was filtered. The filtrates were combined, then concentrated to a density of 1 g of crude drug per milliliter. While being stirred slowly, ethanol was added to the solution until its concentration reached 65%, then allowed to stand for 12 h. Afterwards, the solution was filtered to collect the filtrate, which was concentrated to recycle ethanol. Subsequently, the solution was extracted four times with water-saturated n-butanol, after which n-butanol layers were collected and combined, with n-butanol recycled under reduced pressure. The n-butanol extract was dissolved in 80% ethanol and loaded onto a polyacrylamide column. Elution was performed using 8 times the column volume of 80% ethanol, while the eluate and wash fractions were collected, and ethanol was recovered under reduced pressure. The residue underwent microwave vacuum drying to obtain the *P. orientale* extract (POE) with a yield of 2.62%. In this study, the content of these active constituents is as follows: protocatechuic acid at 0.1623 mg/g, isoorientin at 40.5618 mg/g, orientin at 17.0691 mg/g, vitexin at 45.8365 mg/g, kaempferol-3-O-β-d-glucoside at 0.4784 mg/g, paprazine at 1.4984 mg/g, *N*-*trans*-feruloyltyramine at 7.2463 mg/g, and quercetin at 16.8070 mg/g.

### 4.5. Preparation of Active Ingredient Group Solution (Mix)

Based on the molar concentration ratios of various components in the POE, approximating 1:85:35:10:0:1:5:20:50, a mixture of active constituents from PO (Mix) was prepared with the following doses: 0.1 µmol/L of protocatechuic acid, 8.5 µmol/L of isoorientin, 3.5 µmol/L of orientin, 10.0 µmol/L of vitexin, 0.1 µmol/L of kaempferin-3-O-β-d-glucoside, 0.5 µmol/L of paprazine, 2 µmol/L of *N*-*trans*-phenylethylferoyltyramine, and 5 µmol/L of quercetin.

### 4.6. Preparation of a Mouse Myocardial Ischemia Model

Male ICR mice (25 ± 5 g) free from specific pathogens were procured from Spiff (Beijing) Biotechnology Co., Ltd., with license number SCXK (Jing) 2019-0010. Prior to the commencement of the experiments, all mice underwent a one-week acclimatization period in a controlled environment within a standard laboratory (SPF laboratory), where the environmental conditions were maintained at a temperature range of 20–24 °C, with humidity levels at 55 ± 5%, operating on a 12/12-h light/dark cycle. Mice were provided with ad libitum access to both food and water. All experimental protocols for mice were approved by the Animal Experimentation Ethics Committee of Guizhou Medical University (the approval number is 1801209). The protocols adhered to the regulations of Guizhou Medical University for the management of laboratory animals and the “Animal Protection Law of the People’s Republic of China”.

Anesthetized with 2% tribromoethanol through intraperitoneal injection, the mice were connected to a small animal respirator via non-invasive endotracheal intubation. With a left thoracotomy performed at the 3rd and 4th ribs on the left lateral chest wall, the heart was exposed, and then the left anterior descending branch of the coronary artery was ligated using 0-gauge sutures. Subsequently, the heart was promptly repositioned within the thoracic cavity, while the chest wall was meticulously sutured. Ten minutes later, the mouse’s ECG was monitored using an electrocardiogram (ECG) machine with limb leads. A significant elevation of the ST segment in the two-lead ECG was used as an indicator of the successful ligation.

### 4.7. Animal Grouping and Drug Administration

The mice were randomly divided into the following groups: Sham group, Myocardial Ischemia (MI) group, Myocardial Ischemia + POE (4 g crude drug/kg, POE) group, and Myocardial Ischemia + Metoprolol Tartrate (6.5 mg/kg, MTT) group, with each group comprising 8 mice. Since our research team has previously investigated different dosages of PO for treating cardiovascular diseases across various models, the dosage design for the current study is based on the outcomes of these preliminary experiments [35]. Except for the Sham group, the mice in the remaining groups suffered from myocardial ischemia using the method described above. In the Sham group, a thoracotomy was performed without coronary artery ligation, whereas all other procedures were identical to those in the other groups. Subsequently, the mice with successfully induced myocardial ischemia were randomly divided into three groups, namely MI, POE, and MTT groups, in which the POE and MTT groups received corresponding drug solutions via oral gavage once daily for 14 consecutive days. The Sham group (Sham) and the MI group were administered an equivalent volume of blank solvent.

### 4.8. Electrocardiogram Examination

After the last administration, the mice were anesthetized with 2% tribromoethanol, placed in a supine position, connected to an electrocardiogram machine, and subjected to electrocardiogram monitoring using limb leads. The changes in ST segment waveform were measured, and the amplitude changes were recorded for each mouse.

### 4.9. TTC Staining

After being harvested and rinsed with physiological saline, the mouse hearts were rapidly placed in a −20 °C freezer for 30 min. Subsequently, a 2 mm thick section was sliced from the ligation site to the apex of the heart, placed in a 1% TTC staining solution, incubated at 37 °C in the dark for 15–20 min, and then photographed and documented. Areas appearing gray-white were identified as myocardial infarction zones, in which the area was quantified using Image J software (version no. 1.52) as a percentage of the left ventricle’s total area.

### 4.10. Hematoxylin-Eosin Staining

The cardiac tissues of mice were procured and preserved in 10% formalin for 24 h. Following routine paraffin embedding, sections with a thickness of 5 µm were obtained. The deparaffinization process involved the use of xylene, followed by sequential incubations in varying concentrations of ethanol, concluding with a rinse in distilled water. Hematoxylin staining was carried out for a duration of 5 min, with excess stain removal through water rinsing. Differentiation was accomplished by immersing the sections in hydrochloric acid ethanol for 30 s, followed by a 15-min water bath. Subsequently, eosin staining was performed for 2 min. Dehydration, transparency, and sealing procedures were executed accordingly. Finally, the optical microscope was employed to observe the histopathological changes in the cardiac tissue.

### 4.11. Determination of Myocardial Enzyme and Brain Natriuretic Peptide (BNP)

Blood samples were acquired from the ocular blood vessels of mice within each group. Subsequently, mouse serum was isolated, and the concentrations of creatine kinase (CK) and brain natriuretic peptide (BNP) were assessed in accordance with the kit’s provided instructions.

### 4.12. Cells Culture and Treatment

H9c2 cardiomyocytes obtained from American Type Culture Collection (ATCC, Manassas, VA, USA) and cultured in a medium consisting of 10% FBS, 1% penicillin-streptomycin antibiotic, and 89% high-glucose DMEM at 37 °C with 5% CO_2_. Cells in the exponential phase were utilized for subsequent passages or experiments.

The treatment groups included a normal control group, an H_2_O_2_ group, an active ingredient group (Mix), and a POE (POE 80 µg/mL) group. We investigated the protective effects of different dosages of PO extract on the cell model, with results presented in the Appendix A. Since the current study primarily explores the mechanism of action of PO, subsequent research utilized only the highest dosage. Furthermore, the dosage design for the active component group of PO was also based on the dosage of the PO extract. When reached a confluence of 70–80%, H9c2 cells were incubated with the corresponding drug (POE, Mix), and cells of control and model groups were subjected to the medium without drug. The H_2_O_2_, Mix, and POE groups received 400 µmol/L H_2_O_2_ solution prepared in high-glucose DMEM without FBS for 0.5 h at 37 °C in a 5% CO_2_ incubator. Relevant indexes were determined after incubation.

### 4.13. Cell Viability Assay

H9c2 cells were subjected to trypsinization for dispersion and subsequently seeded into a 96-well plate at a density of 8 × 10^3^ cells per well, allowing for a 24-h incubation period. Following treatment with the designated drugs or H_2_O_2_ solution for each experimental group, cell viability was assessed using a CCK-8 kit. The detection process adhered to the manufacturer’s protocol, and measurements were conducted at 450 nm employing a Varioskan LUX microplate reader (ThermoFisher, Vantaa, Finland).

### 4.14. Assay for LDH, SOD, and CAT

After the experimental treatment, the cell culture medium underwent centrifugation at 2500 rpm for 10 min. The resulting supernatant was carefully collected for the assessment of LDH activity, following the precise protocol outlined by the manufacturer. Following this step, the cells were washed twice with phosphate-buffered saline (PBS) and subsequently lysed using RIPA buffer. After centrifugation, the obtained homogenate was utilized for the determination of total SOD and CAT activity in accordance with the manufacturer’s instructions.

### 4.15. Intracellular ROS Assay

Reactive oxygen species (ROS) were assessed using the fluorescent probe DCFH-DA, following the manufacturer’s guidelines provided in the ROS test kit. Specifically, H9c2 cells, subjected to the aforementioned treatment, were incubated with 10 mM DCFH-DA at 37 °C for 1 h. Subsequently, any surplus DCFH-DA was thoroughly removed with PBS, and the fluorescence intensity was quantified employing a fluorescence microplate reader with an excitation wavelength of 488 nm and emission wavelength of 525 nm.

### 4.16. Assay for Mitochondrial Membrane Potential (mtΔΨ)

With the respective reagents administered to each experimental group, the mitochondrial membrane potential was assessed using the JC-1 assay kit, following the prescribed kit procedure. In brief, H9c2 cells were treated with the JC-1 dye working solution at 37 °C for 20 min in the absence of light and subsequently washed twice with the dye buffer. Subsequently, fluorescence microscope imaging rapidly captured the data, and changes in mitochondrial membrane potential were quantified by determining the ratio of red fluorescence intensity to green fluorescence.

### 4.17. Annexin V-FITC/PI Double Staining

Apoptosis was assessed using Annexin-V-FITC and propidium iodide (PI) staining via C6 plus flow cytometry (BD Biosciences, Franklin Lakes, NJ, USA). Following the designated treatment, H9c2 cells were harvested with trypsin. The collected cells were then resuspended in 1× Annexin V binding buffer (195 μL). Subsequently, Annexin V-FITC (5 μL) and PI (10 μL) were added for cell incubation at room temperature for 15 min in the absence of light. The apoptotic rate was determined through flow cytometry.

### 4.18. Western Blot Analysis

Heart tissues or cells were lysed in ice-cold RIPA lysis buffer, and lysates were collected by centrifugation at (4 °C, 12,000× *g*) for 10 min. The protein concentration was determined using a BCA assay kit. The total protein was then combined with loading buffer, subjected to vortex mixing, heated at 100 °C for 5 min, cooled, packaged, and stored at −80 °C until use. Approximately 30 μg of protein samples were separated on a 10% sodium dodecyl sulfate polyacrylamide gel (SDS-PAGE) and subsequently transferred to a PVDF membrane. The membrane was sealed with 5% BSA and incubated overnight at 4 °C with primary antibodies, including Bax (1:10,000), Caspase 3 (1:1000), Caspase 9 (1:1000), SOD2, AC-SOD2, SIRT3, and GAPDH (1:1000), respectively. Following washing, the membranes were incubated with the corresponding secondary antibody (1:500) for 2 h at room temperature. After washing with TBST, the antigen-antibody complexes were visualized using ECL reagent and quantitatively analyzed employing Quantity One software (version no. V4.6.6, Bio-Rad Laboratories, Hercules, CA, USA).

### 4.19. Reverse Transcription Quantitative Polymerase Chain Reaction (RT-qPCR Analysis)

Upon cellular treatment, H9c2 cell total RNA was extracted using the RNAsimple Total RNA Kit and subsequently converted to cDNA using the PrimeScript™ RT Master Mix kit (Takara Bio Inc., Kusatu, Japan). Quantitative PCR (qPCR) was conducted using the TB GreenR Premix EX TaqTM II kit (Takara Bio Inc.) on a Real-Time PCR detection system. The 2^−ΔΔCt^ method was employed to calculate the relative expression of the target gene, with GAPDH serving as the reference endogenous gene for data normalization. The primer sequences utilized in this study are as follows: GAPDH forward primer: 5′-CAAGAAGGTGGTGAAGCAG-3′, Reverse primer: 5′-CAAAGGTGGAAGAATGGG-3′; SOD2 forward primer: 5′-CGTGACTTTGGGTCTTTTG-3′, Reverse primer: 5′-CGGCAATCTGTAAGCGA-3′.

### 4.20. Statistical Analysis

Data were statistically analyzed using GraphPad Prism 8 and SPSS 22.0 software. Experimental data were expressed as mean ± standard deviation, and for multiple comparisons, one-way analysis of variance was used for comparison between groups. *p* < 0.05 was considered statistically significant.

## 5. Conclusions

Taken together, this study demonstrated the potent protective effects of Mix on myocardial cells through modulation of the SIRT3/SOD2 signaling pathway. Notably, our findings present the novel discovery that the regulation of this pathway is a key mechanism underlying the cardiovascular benefits of PO. This new insight offers a strong foundation for positioning the Mix as a promising candidate for the development of cardiovascular therapies. Nevertheless, as these effects have only been observed in animal and cell models, further research is needed to validate these results in both preclinical and clinical settings, further exploring the therapeutic potential of the Mix in targeted cardiovascular interventions.

## Figures and Tables

**Figure 1 pharmaceuticals-17-01288-f001:**
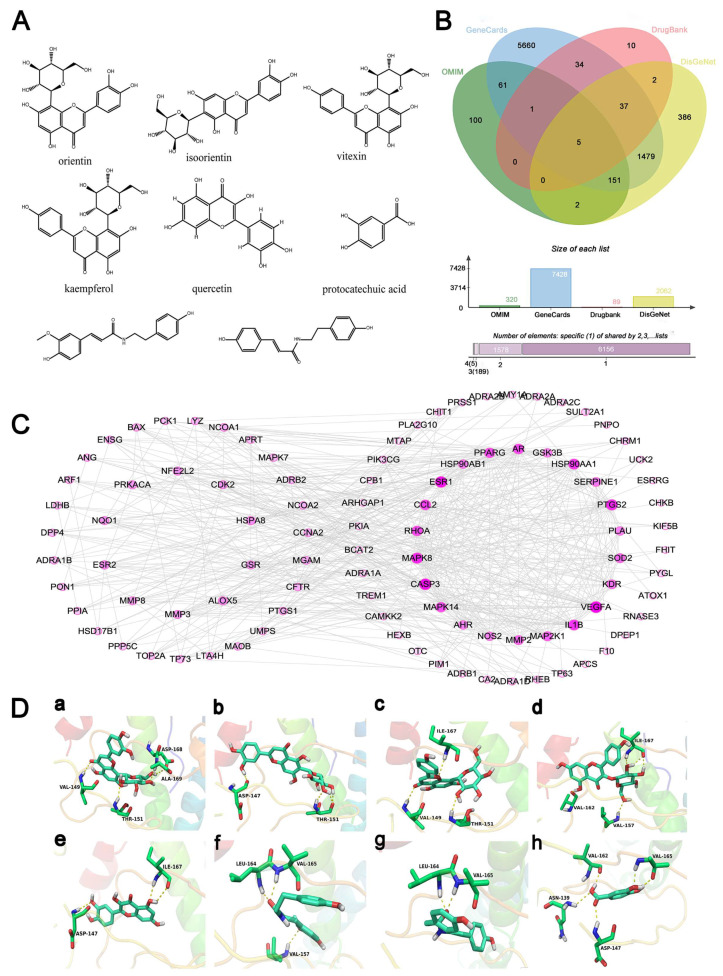
Potential targets of the active components of PO for treating cardiovascular diseases. (**A**) the structures of eight active compounds of PO; (**B**) the genes related to cardiovascular diseases; (**C**) the PPI network of potential targets of the active components for cardiovascular diseases; (**D**) the docking diagram of eight compounds with SOD2: (**a**) docking mode of orientin with SOD2; (**b**) docking mode of isoorientin with SOD2; (**c**) docking mode of vitexin with SOD2; (**d**) docking mode of kaempferol with SOD2; (**e**) docking mode of quercetin with SOD2; (**f**) docking mode of *N*-*trans*-feruloyltyramine with SOD2; (**g**) docking mode of paprazine with SOD2; (**h**) docking mode of protocatechuic acid with SOD2.

**Figure 2 pharmaceuticals-17-01288-f002:**
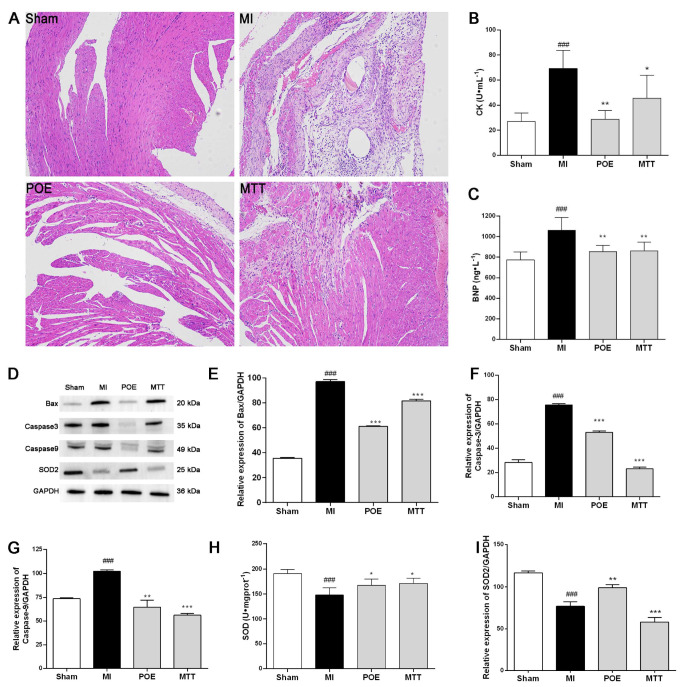
PO extract alleviates the myocardial damage and apoptosis and enhances the antioxidant capacity in myocardial infarction (MI) mice induced by LAD ligation. (**A**) Histological results of heart tissue in mice using hematoxylin-Eosin staining (Scale: 100 μm); (**B**) the CK level in serum samples; (**C**) the BNP level in serum samples; (**D**) The protein expressions of Bax, Caspase 3, Caspase 9, and SOD2 in heart tissue assessed using Western blotting analysis; (**E**–**G**) Quantification of Bax (**E**), Caspase 3 (**F**), Caspase 9 (**G**); (**H**) The activity of SOD in heart tissue; (**I**) Quantification of SOD2. ### *p* < 0.001, vs. the Sham group; * *p* < 0.05, ** *p* < 0.01, *** *p* < 0.001, vs. the MI group.

**Figure 3 pharmaceuticals-17-01288-f003:**
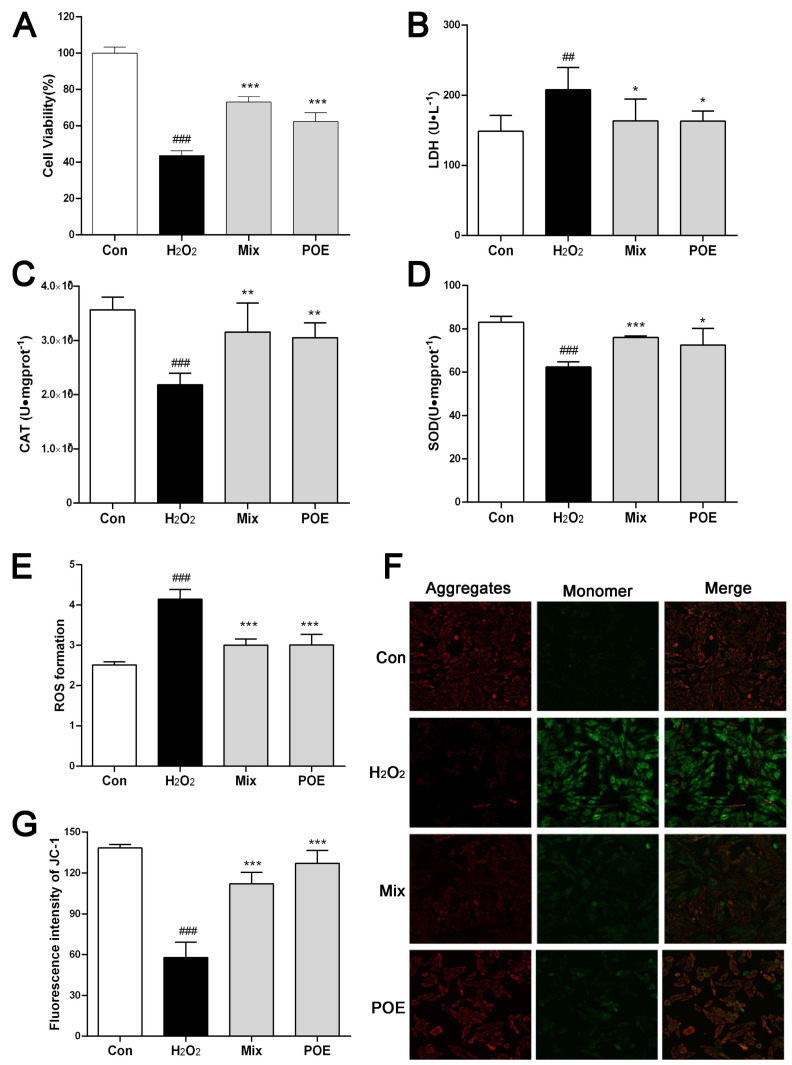
The active component mixture (Mix) from PO effectively protects H9c2 cardiomyocytes and mitochondria from injury caused by H_2_O_2_, enhancing antioxidant capacity. (**A**) The effect of POE and Mix on the cell viability assessed by CCK-8 kits; (**B**) effect of POE and Mix on LDH activity in the cell culture medium; (**C**,**D**) effect of POE and Mix on CAT (**C**) and SOD (**D**) activity in H9c2 homogenate; (**E**) effect of POE and Mix on ROS level using the fluorescent probe DCFH-DA; (**F**) effect of POE and Mix on mitochondrial membrane potential evaluated by a JC-1 assay kit (Scale: 100 μm); (**G**) quantification of fluorescence intensity ratio of red fluorescence to green fluorescence. ## *p* < 0.01, ### *p* < 0.001, vs. the Con group; * *p* < 0.05, ** *p* < 0.01, *** *p* < 0.001, vs. the H_2_O_2_ group.

**Figure 4 pharmaceuticals-17-01288-f004:**
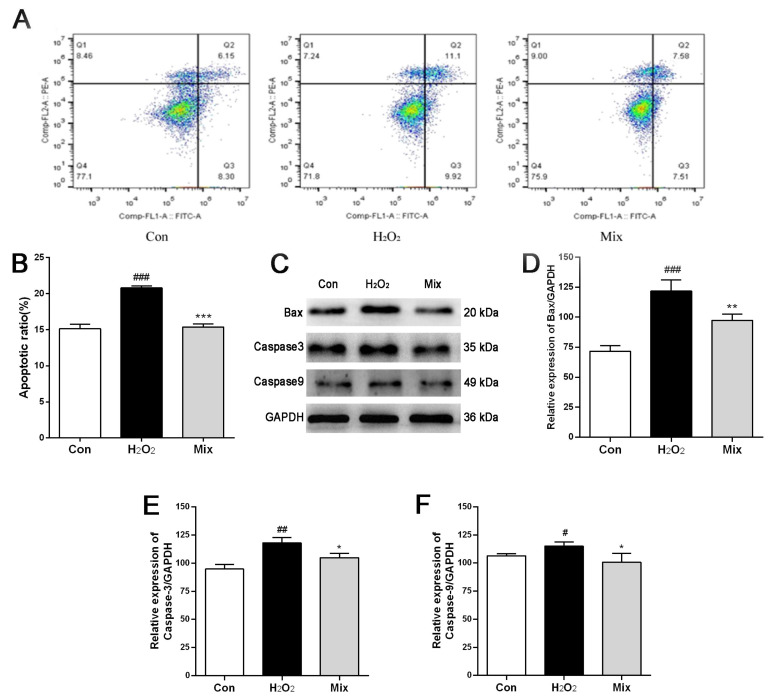
The active component mixture (Mix) from PO mitigates cell apoptosis induced by H_2_O_2_. (**A**) Effect of Mix on apoptosis rate assessed using Annexin V-FITC/PI double staining; (**B**) statistical diagram of apoptosis rate; (**C**) protein expression of Bax, Caspase 3, and Caspase 9 was detected by Western blotting analysis; (**D**–**F**) quantitation of Bax (**D**), Caspase 3 (**E**), and Caspase 9 (**F**). # *p* < 0.05, ## *p* < 0.01, ### *p* < 0.001, vs. the Con group; * *p* < 0.05, ** *p* < 0.01, *** *p* < 0.001, vs. the H_2_O_2_ group.

**Figure 5 pharmaceuticals-17-01288-f005:**
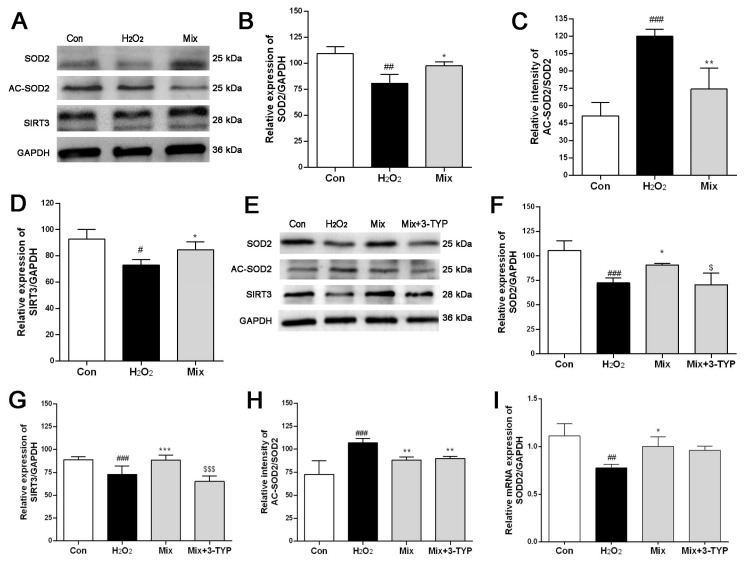
Effect of active component mixture (Mix) on the SIRT3/SOD2 signaling pathway. (**A**) Effect of Mix on the protein expression of SOD2, AC-SOD2, and SIRT3 detected by Western blotting analysis; (**B**–**D**) quantitation of SOD2, AC-SOD2, and SIRT3; (**E**) effect of Mix on the protein expression of SOD2, AC-SOD2, and SIRT3 was regulated by the inhibitor 3-TYP of SIRT3; (**F**–**H**) quantitation of SOD2, AC-SOD2, and SIRT3 after pre-treatment with 3-TYP; (**I**) the mRNA level of SOD2 was affected by Mix and 3-TYP. # *p* < 0.05, ## *p* < 0.01, ### *p* < 0.001, vs. the Con group; * *p* < 0.05, ** *p* < 0.01, *** *p* < 0.001, vs. the H_2_O_2_ group; $ *p* < 0.05, $$$ *p* < 0.001, vs. the Mix group.

**Figure 6 pharmaceuticals-17-01288-f006:**
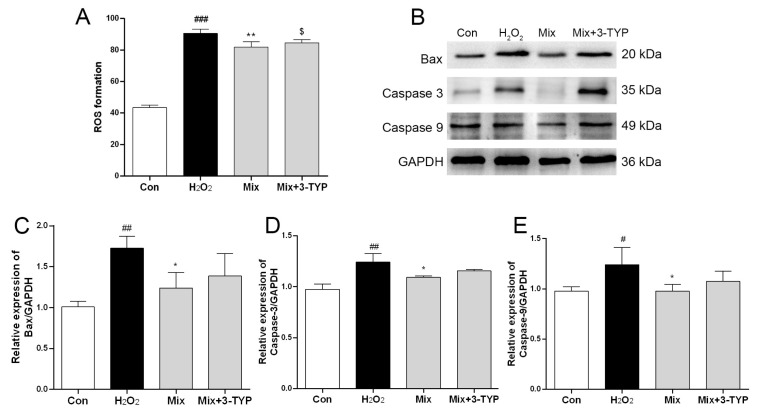
Involvement of SIRT3/SOD2 signaling pathway in the cardioprotective effect of Mix. (**A**) ROS level detected by the fluorescent probe DCFH-DA; (**B**) protein expression of Bax, Caspase 3, and Caspase 9 was detected by Western blotting analysis; (**C**–**E**) quantitation of Bax (**C**), Caspase 3 (**D**), and Caspase 9 (**E**). # *p* < 0.05, ## *p* < 0.01, ### *p* < 0.001, vs. the Con group; * *p* < 0.05, ** *p* < 0.01, vs. the H_2_O_2_ group; $ *p* < 0.05, vs. the Mix group).

**Figure 7 pharmaceuticals-17-01288-f007:**
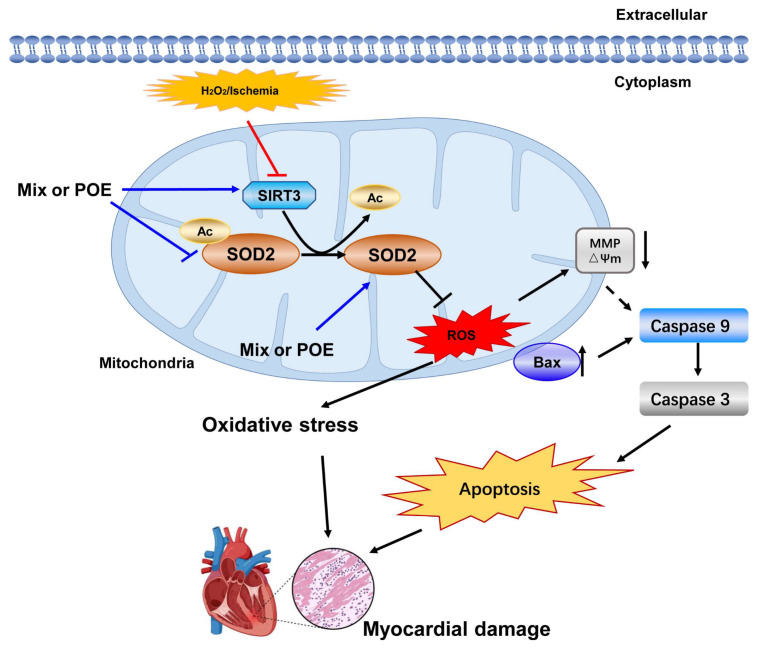
Molecular mechanism of Mix for treatment of myocardial damage.

**Table 1 pharmaceuticals-17-01288-t001:** Binding energy (kcal/mol) of eight active ingredients to the top twenty key targets.

	Orientin	Isoorientin	Vitexin	Kaempferol-3-O-β-d-glucoside	Quercetin	*N*-*Trans*-feruloyltyramine	Paprazine	Protocatechuic-Acid
AR	−5.387	−5.93	−5.579	-	−9.634	−6.781	−8.055	−7.275
ESR1	−5.049	−4.402	−5.062	−4.633	−6.551	−4.212	−2.489	−5.168
HSP90AA1	−3.741	-	−4.498	−4.631	-	−3.179	−3.072	−5.062
HSP90AB1	−4.075	-	−3.975	−4.55	−4.473	−3.471	−1.433	−4.676
IL1B	−5.349	−6.33	−4.721	−6.143	−5.719	−5.834	−4.54	−5.164
KDR	−5	−5.026	−4.634	−4.574	−5.476	−3.785	−3.395	−4.005
MAP2K1	-	-	-	-	−6.588	−6.242	−3.108	-
MAPK8	−3.082	−2.939	−3.411	−3.377	−3.323	−2.974	−2.231	−4.288
MAPK14	−4.405	−4.357	−4.248	−3.875	−4.726	−4.193	−3.715	−4.856
MMP2	−3.622	-	−3.116	-	−3.97	−3.168	−1.874	−4.531
PLAU	−4.186	−4.759	−4.657	−4.887	−5.961	−3.896	−2.202	−4.468
PPARG	−6.605	−5.914	−7.555	−6.723	−6.519	−8.264	−5.965	−5.571
PTGS2	−4.418	−5.238	−4.523	−3.867	−5.643	−5.104	−2.203	−4.609
RHOA	−3.806	−5.305	−2.942	−3.251	−5.045	−4.771	−3.557	−4.708
SERPINE1	−3.031	−3.805	−3.544	−3.449	−4.376	−3.046	−2.08	−3.982
AHR	−5.723	−6.754	−5.925	−6.152	−6.719	−6.984	−4.45	−5.756
CASP3	−5.45	−5.245	−4.77	−5.433	−4.936	−4.927	−4.348	−4.947
CCL2	−5.866	−6.015	−7.127	−5.607	−5.774	−5.846	−5.942	−6.57
SOD2	−7.468	−6.831	−7.258	−6.818	−6.617	−6.561	−5.162	−6.666
VEGFA	−4.668	−4.152	−4.468	−4.305	−4.933	−3.327	−2.695	−4.616

Note: ‘-’ indicates the absence of binding between receptors and ligands.

**Table 2 pharmaceuticals-17-01288-t002:** Changes in ST amplitude and the size of myocardial infarction in each group.

Group	ST Segment	Myocardial Infarction Area (%)
Sham	0.59 ± 0.04	12.37 ± 0.29
MI	0.05 ± 0.03 ***	39.79 ± 5.38 ***
POE	0.23 ± 0.02 ^###^	26.21 ± 2.51 ^#^
MTT	0.26 ± 0.04 ^###^	21.82 ± 4.28 ^#^

Note: *** *p* < 0.001 vs. Sham group, ^#^
*p* < 0.05, ^###^
*p* < 0.001 vs. MI group.

## Data Availability

All the relevant data are provided within the paper, and data in the current study are available from the corresponding author.

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
