# Peer review of "Regulation of the SIRT3/SOD2 Signaling Pathway by a Compound Mixture from Polygonum orientale L. for Myocardial Damage"

_pharmaceuticals, 2024, doi:10.3390/ph17101288_

Round 1

Reviewer 1 Report

Comments and Suggestions for Authors

The present study titled “Regulation of the SIRT3/SOD2 Signaling Pathway by a Compound Mixture from Polygonum orientale L. for Myocardial Damage” presents a comprehensive theoretical and experimental investigation into the cardioprotective effects of compounds from Polygonum orientale L (OP). In silico techniques, primarily based on network pharmacology and molecular docking, were employed to identify potential biological targets. Additionally, in vitro and in vivo studies were conducted to evaluate the cardioprotective efficacy and mechanisms of action of OP and its active ingredients. Overall, the subject is of interest, and the paper is well-written and organized. Therefore, it is suitable for publication. However, there are some suggestions and corrections to be considered before publication.

- On page 3, lines 109-110, references need to be added to support this statement.

- The resolution of Figure 1 is very low; a high-resolution version of the figure needs to be added.

- There are some gaps in Table 1; please review and correct them.

- On page 3, lines 137-139, please adjust the text. Figure 1D is not related to binding affinity. Add a sentence indicating that the binding modes of the compounds to the enzyme are shown in Figure 1D.

- Based on the docking results, AR, PPARG, and SOD2 are all potential targets. It does not appear that SOD2 is better than the others. Please revise the paragraph to indicate that SOD2 was selected as it is one of the most potential targets.

Author Response

comments1: On page 3, lines 109-110, references need to be added to support this statement.

Response1: Thank you for bringing this to our attention. We have added the appropriate references to support the statement on page 3, lines 109-110.

comments2: The resolution of Figure 1 is very low; a high-resolution version of the figure needs to be added.

Response 2: Thank you for your feedback; we will re-upload the high-resolution Figure 1.

comments3: There are some gaps in Table 1; please review and correct them.

Response 3: Thank you for pointing this out. We have reviewed and corrected the gaps in Table 1, and added the corresponding annotations.

comments4: On page 3, lines 137-139, please adjust the text. Figure 1D is not related to binding affinity. Add a sentence indicating that the binding modes of the compounds to the enzyme are shown in Figure 1D.

Response 4: Thank you for your suggestion. We have added the expression in our revised manuscript.

comments5: Based on the docking results, AR, PPARG, and SOD2 are all potential targets. It does not appear that SOD2 is better than the others. Please revise the paragraph to indicate that SOD2 was selected as it is one of the most potential targets.

Response 5: Thank you for your insightful comments. We have revised the paragraph to clearly explain the reason why SOD2 was selected. Specifically, oxidative damage plays a crucial role in the pathology of cardiovascular diseases, and there was evidence that PO and/or these compounds have demonstrated significant antioxidant effects in their treatment. As SOD2 is a key enzyme in the body's antioxidant system, in this study SOD2 was selected as a focal point for further investigation.

Reviewer 2 Report

Comments and Suggestions for Authors

The MS titled “Regulation of the SIRT3/SOD2 Signaling Pathway by a Compound Mixture from Polygonum orientale L. for Myocardial Damage” by Chunhua Liu et al., highlighted the probable mechanisms of Polygonum orientale extract (POE) for having cardioprotective properties, and reported that the involvement of SIRT3/SOD2 signaling pathway.

Overall the study looks impressive but the authors used plant extract (eight bioactive metabolite/molecules) rather than single molecule, so it’s confusing/not clear which active molecule is responsible for the claimed activity or the activity might be due to synergistic effects, or even sometime in a mixture they have inhibitory effects on each other. So better to conclude/identify and summarize that which active molecule is responsible for the cardioprotective potential. Further, after identification make it clear whither it is the first time to report the said activity.

Comments on the Quality of English Language

Need a good review of the language and grammar throughout the text. 

Author Response

comments1: Overall the study looks impressive but the authors used plant extract (eight bioactive metabolite/molecules) rather than single molecule, so it’s confusing/not clear which active molecule is responsible for the claimed activity or the activity might be due to synergistic effects, or even sometime in a mixture they have inhibitory effects on each other. So better to conclude/identify and summarize that which active molecule is responsible for the cardioprotective potential. Further, after identification make it clear whither it is the first time to report the said activity.

Response1: Thank you for your insightful feedback. Indeed, it is challenging to pinpoint which specific component is responsible for the therapeutic effect when working with plant extracts containing multiple constituents. However, traditional Chinese medicine is characterized by the synergistic action of multiple components. Therefore, in this study, we focused on a group of bioactive compounds that were preliminarily identified in previous research. The objective of this study is to determine the efficacy of this group of active compounds. In future research, we will conduct an in-depth investigation into the individual effects of each of the eight compounds, as well as their synergistic interactions.

comments2: Need a good review of the language and grammar throughout the text. 

Response2:Thank you for your comments! We apologize for the language problems in the manuscript. The grammar of this manuscript has been revised with assistance from a native English speaker with appropriate research background.

Reviewer 3 Report

Comments and Suggestions for Authors

Authors suggested that Mix demonstrates a substantial protective effect on myocardial cells via modulation of the SIRT3/SOD2 signaling pathway. This research innovatively identifies, for the first time, the regulation of the SIRT3/SOD2 signaling pathway as one of new critical mechanisms by which PO exerts its anti-cardiovascular disease effects. Figures/Images are of good standard and are clear enough to understand the results of the manuscript. Methodology is well drafted and organized. Introduction section is well elaborated with sufficient number of citations. Research seems to be novel as it has provided a potential drug candidate for the development of treatments for cardiovascular diseases. However i would like to suggest the potential author to check for typological error and rewrite the conclusion section emphasizing more on the outcome and futuristic aspects if the research study. Hence, i would recommend the article to be accepted in its presented form with some minor changes.

Comments on the Quality of English Language

English is fine. Check for typological errors

Author Response

comments1: I would like to suggest the potential author to check for typological error and rewrite the conclusion section emphasizing more on the outcome and futuristic aspects if the research study. 

Response1: Thank you for your comments. We have made adjustments to the conclusion section, focusing more on the research findings and future prospects.

Reviewer 4 Report

Comments and Suggestions for Authors

Overall Evaluation: This manuscript provides an in-depth exploration of the cardioprotective mechanisms of Polygonum orientale L., with a particular focus on its regulation of the SIRT3/SOD2 signaling pathway. The research topic is of significant clinical and scientific value, addressing a crucial area in cardiovascular disease treatment. The study is methodologically rigorous, utilizing advanced techniques such as network pharmacology, molecular docking, and in vivo and in vitro experiments. The findings are robust and well-supported by the data, and the study's innovation and depth make it a valuable contribution to the field of cardiovascular research. Notably, the study is the first to reveal the critical role of the SIRT3/SOD2 signaling pathway in the cardioprotective effects of Polygonum orientale, opening new avenues for therapeutic approaches in cardiovascular diseases.

Areas for Improvement: I recommend this manuscript for publication after addressing a few points. The discussion could benefit from more comprehensive literature citations, including references to the latest studies, to enhance the manuscript's relevance and scientific rigor. Additionally, the discussion section should be expanded to explore other potential molecular mechanisms that Polygonum orientale might involve and compare these findings with existing cardiovascular treatments. Furthermore, the language and terminology could be simplified or clarified to ensure broader accessibility, and some sections could be made more cohesive for smoother reading. The current study bases its experiments on the ratios of components in the extract, but these ratios might not represent the optimal combination. I suggest conducting dose-response studies to optimize the combination ratios of the active components and explore potential synergistic or interactive effects between different components, which could reveal additional mechanisms and therapeutic potential. This will help develop more effective treatment strategies and may enhance the clinical efficacy of the PO mixture.

Author Response

comments1: The discussion could benefit from more comprehensive literature citations, including references to the latest studies, to enhance the manuscript's relevance and scientific rigor. Additionally, the discussion section should be expanded to explore other potential molecular mechanisms that Polygonum orientale might involve and compare these findings with existing cardiovascular treatments. Furthermore, the language and terminology could be simplified or clarified to ensure broader accessibility, and some sections could be made more cohesive for smoother reading. The current study bases its experiments on the ratios of components in the extract, but these ratios might not represent the optimal combination. I suggest conducting dose-response studies to optimize the combination ratios of the active components and explore potential synergistic or interactive effects between different components, which could reveal additional mechanisms and therapeutic potential. This will help develop more effective treatment strategies and may enhance the clinical efficacy of the PO mixture.

Response1: Thank you for your valuable suggestions, which we fully agree with. We have updated the references to improve the manuscript's relevance and expanded the discussion section accordingly. Additionally, the language has been refined for greater clarity and accessibility.

Regarding the component ratios used in our study, these were based on the natural proportions found in the extract and previous preliminary studies. However, we agree that these may not represent the optimal combination. In future research, we plan to conduct dose-response studies to optimize the ratios of the active components. This will allow us to further investigate potential synergistic or interactive effects between different components, which could reveal additional mechanisms and enhance the therapeutic potential of Polygonum orientale. Such efforts will aid in developing more effective treatment strategies and may improve the clinical efficacy of PO-based therapies.

Round 2

Reviewer 2 Report

Comments and Suggestions for Authors

Its fine but being in this form the level of novelty is poor and unclear.

Comments on the Quality of English Language

Improved now

Author Response

comments1: Its fine but being in this form the level of novelty is poor and unclear.

Response1: Thank you for your valuable suggestions. Using extracts and mixtures to study their activity is indeed a relatively common method. Importantly, our research has, for the first time, revealed that Polygonum orientale L. can exert a protective effect on cardiomyocytes by regulating the Sirt3/SOD2 pathway.  In future studies, we will conduct in-depth research on the individual components of the active ingredient group of Polygonum orientale L. and their synergistic effects, further enhancing our innovation.